# Assessing Forest Quality through Forest Growth Potential, an Index Based on Improved CatBoost Machine Learning

**Lianjun Cao** [1,2,3], **Xiaobing He** [4], **Sheng Chen** [5] **and Luming Fang** [1,2,3,*]

1 College of Mathematics and Computer Science, Zhejiang A&F University, Hangzhou 311300, China; cn2378@stu.zafu.edu.cn
2 Key Laboratory of State Forestry and Grassland Administration on Forestry Sensing Technology and Intelligent Equipment, Hangzhou 311300, China
3 Key Laboratory of Forestry Intelligent Monitoring and Information Technology of Zhejiang Province, Hangzhou 311300, China
4 Baishanzu Scientific Research Monitoring Center, Qianjiangyuan-Baishanzu National Park, Lishui 323000, China
5 Zhejiang Forest Resources Monitoring Center, Hangzhou 311300, China
* Correspondence: flming@zafu.edu.cn

**Abstract:** Human activities have always depended on nature, and forests are an important part of this; the determination and improvement of forest quality is therefore highly significant. Currently, domestic and foreign research on forest quality focuses on the current states of forests. We propose a new research direction based on the future states. By referencing and analyzing the forest quality standards of domestic and foreign experts and institutions, the concept and model for calculating forest growth potential were constructed. Forest growth potential is a new forest quality indicator. Based on the data of 110,000 subcompartments of forest resources from the Lin'an and Landsat8 satellites' remote sensing data, the unit volume was predicted using three machine-learning algorithms: random gradient descent SGD, the integrated machine learning algorithm CatBoost, and deep learning CNN. The CatBoost algorithm model was improved based on Optuna; then the improved CatBoost algorithm was selected through evaluation indicators for the prediction of forest volume and finally incorporated into the calculation model for forest growth-potential value. The forest growth-potential value was calculated, and an accurate forest quality improvement scheme based on the subcompartments is preliminarily discussed. The successful calculation of forest growth potential values has a certain reference significance, providing guidance for accurately improving forest quality and forest management. The improved CatBoost calculation model is effective in the prediction of forest growth potential, and the determination coefficient $R^2$ reaches 0.89, a value that compares favorably with those in other studies.

**Keywords:** forest growth potential; precise improvement of forest quality; CatBoost; SGD; CNN

## 1. Introduction

Throughout the history of human development, we have always relied on nature. If human beings want to achieve harmonious and sustainable development, they must understand and correctly handle the dialectical and unified relationship between humans and nature [1]. As an indispensable part of nature, forests are highly important components of terrestrial ecosystems. The forest is a major carbon sink within the terrestrial ecosystem, serving as a crucial mechanism for mitigating the greenhouse effect. Furthermore, the forest functions as a vital reservoir for terrestrial resources, including water and soil conservation. As an ecological barrier, the forest serves an essential role in curtailing the adverse effects of sandstorms and minimizing land desertification [2].

Forest quality is a topic of concern not only for developing countries but for the whole world. For instance, Valero, Picos, and others evaluated the quality indicators and condition of riverbank forests of a section of the Umia River. A total of 55 sampling stations were

evaluated and different performance levels were defined. In total, 64% of the riverbank areas were found to require restoration actions, and over 18% require more extensive protection [3]. Jonne et al. proposed a Bayesian optimization model based on machine learning algorithms using a dataset from the Oran region of Finland, creating a forest volume estimation model and using it to carry out predictions [4]. Forest biomass and carbon monitoring play a crucial role in mitigating climate change. Furthermore, Jukka et al. used Sentinel-2 composite images and process-based ecosystem models to generate estimation and prediction models. The study area includes the boreal forests of Finland and Russia, which are used to predict forest stock in large areas [5].

The fifth national forest resource inventory showed that the area of forests in China was 159 million hectares, and the national coverage rate reached 16.55% [6]. The eighth national forest resource inventory showed that the national forest area reached 208 million hectares, and the forest coverage was 21.63% [7]. The ninth national forest resource inventory showed that the forest coverage rate was 22.96%, the area was 220 million hectares, the stock volume was 17.56 billion cubic meters, and the unit forest-stock volume was 5.32 cubic meters per mu, lower than the world average unit forest-stock volume of 7.19 cubic meters per mu. Meanwhile, the per capita forest-stock volume was 12.35 cubic meters, which is only one-sixth of the world average per capita forest stock volume [6–8]. The survey data of each period show that the Chinese forest area continues to expand and coverage continues to increase, but the quality is relatively low. We propose a conceptualization and mathematical expression of forest growth potential based on our research focus on the future states of forests; we also refer to and analyze the forest quality standards of domestic and foreign experts and institutions. Based on Secondary Resource Surveys of forest subcompartment data and Landsat 8 satellite remote sensing data, we improved the CatBoost algorithm model and selected the improved version to predict the forest volume through certain evaluation indicators. Based on the predicted volume, the model and mathematical expression of forest growth potential are iterated and the forest growth potential value is calculated. The level of forest growth potential reflects the future forest quality, which is, to a certain extent, based on subcompartments. This study therefore produces relatively forward-looking and accurate guidance for improving forest quality, contributing to the goals of carbon peaking and carbon neutrality [9,10] and realizing the preliminary exploration and research into improving forest quality based on a CatBoost algorithm improved by subcompartments.

### 1.1. Forest Quality Indicators

A prerequisite for improving forest quality is the availability of corresponding quality indicators or standards. For example, due to the hierarchical properties of the forest itself, indicators at different scales, such as region, landscape, and stand, have been proposed as sustainable management criteria for forests in the relevant research literature [11,12]. Moreover, due to different researchers focusing on different values or functions, forest quality evaluation indicators for different management objectives have also been generated. These include landscapes, forest management units, and indicators at the regional and the forest stand levels [13].

### 1.2. Traditional Forest Quality Indicators

The 2020 Global Forest Resources Assessment Report included forest areas used for different purposes (timber, water and soil conservation, biodiversity conservation, multifunctional, etc.), forest production capacity (stock volume, carbon storage), damaged forest areas, sustainable forest areas, and forest-related social and economic conditions [14]. In the eighth forest resource inventory, China selected the following as forest quality indicators: the forest stock per hectare, the annual growth per hectare, the number of trees per hectare, the average diameter at breast height, the proportion of nearly mature and over-mature forests, and the proportion of mixed forests, as well as the total biomass and carbon reserves of the national forest vegetation, the annual water conservation, the annual soil fixation, the annual

fertilizer conservation, the annual absorption of pollutants, the annual dust retention, and other ecological functions [15]. Feng, Wang, and others put forward further indicators of forest resource quality, including quantity, structure, productivity, and health [16].

For stand-level indicators, the construction standards of forests for ecological public welfare issued by the State Forestry Administration clearly present the quality evaluation indicators and methods to be used. The standard proposes evaluating water and soil conservation forests, water conservation forests, and other protective forests by means of five indicators: species diversity, canopy density, community level, vegetation coverage, and the litter layer. The forest belt width, integrity, and structure were used to evaluate the stand quality of windbreak and sand-fixation forests, farmland shelterbelts, and other shelterbelts [17]. Based on a comprehensive consideration of various factors, Wang, Zhang, Tong, and others constructed a model for forest quality evaluation through a constraint layer (the forest structure, productivity or economic value, and succession or renewal trend) and an indicator layer (the canopy density, species richness, evenness, thickness of dead branches and leaves, proportion of dead trees, tree stock, shrub and grass biomass, number of seedlings, ratio of dominant young tree seedlings to tree numbers, and seedling mortality) [18]. A summary of these five domestic and foreign mainstream forest quality evaluation indicators is given in Table 1.

**Table 1.** Overview of forest quality evaluation indicators.

| Source of Forest Quality Indicators | Year | Main Indicators of Forest Quality |
|---|---|---|
| Global Forest Resources Assessment [14] | 2020 | Production capacity (storage capacity, etc.)<br>Forest area (timber forest, etc.)<br>Damaged forest area<br>Sustainably managed forest area<br>Forest-related socio-economic conditions |
| National Forest Resources Inventory [15] | 2014 | Stock volume per hectare<br>Average annual growth per hectare<br>Number of plants per hectare, average DBH<br>Total biomass of forest vegetation in nation<br>Carbon storage, solid soil quantity, etc. |
| Feng J G, etc. [16] | 2016 | Quantity<br>Structure<br>Productivity<br>Healthy |
| State Forestry Administration [17] | 2001 | Biodiversity<br>Crown density of forest stand<br>Community level<br>Vegetation coverage<br>Litter layer |
| Wang Naijiang, etc. [18] | 2010 | Forest structure<br>Productivity or economic value<br>Succession or renewal trend<br>Crown density<br>Arbor stock volume |

### 1.3. Construction of a Forest Growth-Potential Index

According to the discussion in Section 1.2, forest stock is an important indicator of forest quality and a basic general indicator. After demonstrating the importance of forest stock through actual forest quality indicators, we offer a brief analysis of its theoretical basis. From an ecological perspective, the main theory is the forest disturbance theory, which refers to discrete events that disrupt the ecosystem or species structure, triggering a change in the effectiveness of the substrate and the environment. Another theory is that of forest succession, which focuses on the process by which one forest community is replaced by another in a given environment. Finally, there is the dominant factor theory.

A dominant factor, also referred to as a limiting factor, is an ecological factor that plays a limiting role in the survival and development of organisms [19]. In addition to the ecological perspective, there is the forestry perspective, which is mainly concerned with modern forestry operations [18].

Forest stock volume remains an important basic indicator in the theoretical basis for improving forest quality. Increasing the amount of accumulation requires increasing the rates of forest cover and growth. To establish new indicators, the general principles and directions refer to those of the basic standard system of the ecological environment described by Chen, Lin, and Li. In a theoretical discussion that establishes a system of basic standards for the ecological environment in China, these writers present the four principles of wholeness, serviceability, universality, and scientificity [20]. Therefore, the increased proportion of the forest unit stock in the future unit time (10 years) is defined as the growth potential. The greater the growth potential, the greater the future increase in the forest stock. This metric not only includes important indicators of forest quality but can also integrate various indicators.

The forest growth-potential value is not directly predicted by the machine-learning model but is nested within the formula of a certain machine-learning model. To simplify the research process, the machine-learning algorithm is introduced separately. The first step is to introduce the machine-learning model, assuming that the improved CatBoost machine-learning model is optimal. The second step is to substitute the calculation formula for the forest growth-potential value for the overall expression.

CatBoost is a machine-learning library opened by the Russian search giant Yandex in 2017; it has become one of the three mainstream artifact libraries of gradient-boosting decision trees (GBDTs) together with lightGBM and xgBoost [21–23]. It has a GBDT framework based on the oblivious trees algorithm, which has few parameters, supports categorical variables, and has high accuracy. Its main purpose is to deal with categorical features efficiently and reasonably. Furthermore, a new method is proposed to deal with the problems of gradient bias and prediction shift to improve the accuracy and generalization ability of the algorithm.

With Pgrow representing the growth potential, $V_{cat}^{y+n}$ representing the unit stock volume derived by the CatBoost algorithm after adding $n$ years to the data year, and $V_{cur}$ representing the current value of the unit stock volume, the forest growth-potential index $P_{grow}$ is calculated as Equation (1):

$$P_{grow} = \frac{V_{cat}^{y+n} - V_{cur}}{V_{cur}} * 100\%$$ (1)

*1.4. General Introduction*

In this paper, the fields of forestry and computer technology are combined. Based on more than 100,000 forest subcompartments of the Secondary Resources Survey and satellite band data from Lin'an, we compare the random gradient descent SGD, the integrated machine-learning algorithm CatBoost, and a deep-learning CNN algorithm. Ultimately, the integrated machine-learning algorithm CatBoost is employed. The unit time is determined as 10 years. The forest survey cycle commonly used at home and abroad is 5 to 10 years. Forest intervention is subject to various resource conditions. In the short and medium terms, 10 years is an ideal time. The actual growth potential is shown in Formula (2), and the process of calculating the forest growth potential is shown in Figure 1:

$$P_{grow} = \frac{V_{cat}^{y+10} - V_{cur}}{V_{cur}} * 100\%$$ (2)

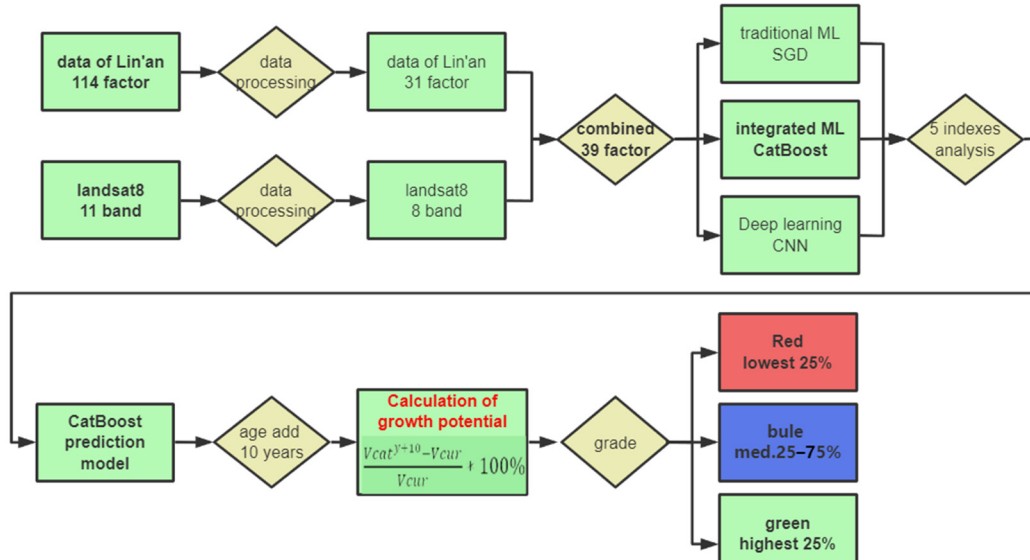

**Figure 1.** Process of calculating growth potential.

The definitions of the short and medium terms can be flexibly applied according to different needs. We are using 10 years in this paper because, according to experience, a comprehensive survey of forestry usually spans 5–10 years, and the change in the forest unit stock is relatively small over a short period. We also need to consider the social benefits but are limited by resources, such as the imbalance between forest coverage and forestry management personnel; it is impossible to make a management adjustment for each subcompartment in a short time.

## 2. Materials and Methods

### 2.1. Research Area and Data Source

The research area is Lin'an District in Hangzhou City, Zhejiang Province. The Lin'an District is situated at the southernmost point of the subtropical monsoon climatic zone in the northwest of Zhejiang Province. Hills and mountains dominate the landscape, which has a clear stereo climate, i.e., there are distinct types of climates distributed simultaneously in the same locations and at the same times. The annual average temperature drops from 16 °C to 9 °C between Jincheng, which is less than 50 m above sea level, and Tianmu Peak, which is 1506 m above sea level, and the annual temperature difference is 7 °C, equivalent to crossing the subtropical and temperate climate zones [24]. The location of the study area as shown in Figure 2.

The primary data source for this analysis is the revised data from the Lin'an District forestry Secondary Survey conducted in 2019. The data source contains 114 columns and 119,792 rows of data records, as shown in Table 2. The other data source is satellite data from the Landsat8 satellite, a new-generation satellite launched by NASA. In this paper, the 6 bands of the OLI imager used are band 2, band 3, band 4, band 5, band 6 and band 7, and the two bands of the TIRS imager are band 10 and band 11 [25]. Anconda2022 is selected as the integrated algorithm platform; the TensorFlow2.0 framework under the integrated platform is used as the framework, and Python version 3.9 is selected for code editing.

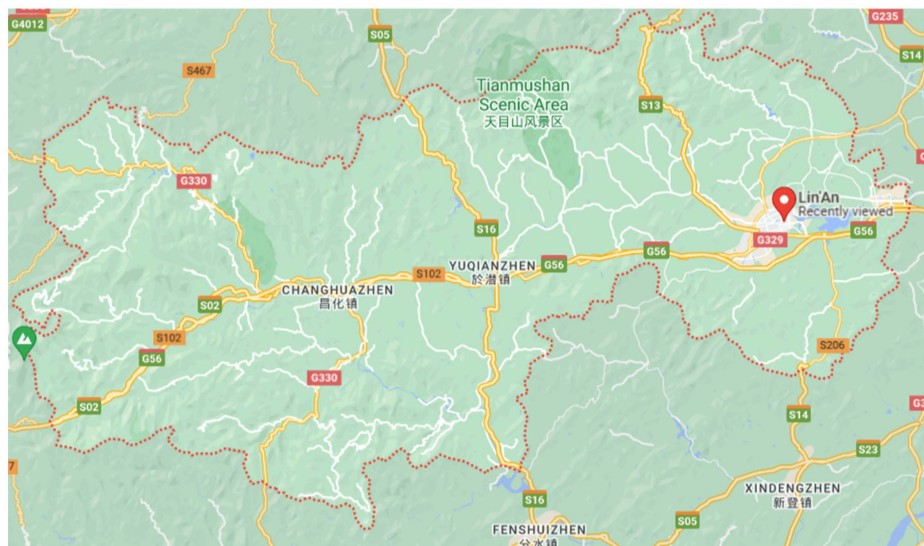

**Figure 2.** The location of the study area.

**Table 2.** Sample data from the Secondary Resources Survey in Lin'an.

| No. | Average DBH (cm) | Density | Average Height (m) | Plants per mu (Plants) | Crown Density | Age | Soil | Vegetation Height (m) |
|---|---|---|---|---|---|---|---|---|
| 0 | 15.3 | 0.51 | 10.1 | 54 | 0.8 | 33 | red | 2 |
| 1 | 15.3 | 0.42 | 10.1 | 45 | 0.7 | 33 | red | 2 |
| 2 | 0 | 0 | 1.2 | 200 | 0.7 | 4 | red | 0.6 |
| 3 | 14.3 | 0.51 | 9.1 | 58 | 0.8 | 31 | red | 2 |
| 4 | 17.3 | 0.51 | 11.1 | 43 | 0.8 | 38 | red | 2 |
| 5 | 9.3 | 0.18 | 6.1 | 37 | 0.4 | 18 | red | 2 |
| … | … | … | … | … | … | … | … | … |

### 2.2. Data Preparation

Data and algorithms are both important and need to be processed to guarantee data availability while retaining as many data as possible to improve the prediction accuracy of the model.

First, the SHP format data from the forestry Secondary Resources Survey were extracted and converted to a CSV database file format, containing 114 factors and 1 target as unit accumulation. Then, the data were cleaned in various ways to identify the invalid factors, removing columns that were clearly unrelated to the experiment, columns that contained many textual descriptions, and columns that could not be quantified. Finally, the data were processed by Pandas to ensure that 95% of the data of any factor were not null, containing meaningless special symbols, garbled codes, etc. [26]. There are also treatments for some data values that exceed reasonable bounds, and the very few over-bound data were also kept so that the model would be more stable and the results would not fluctuate due to incorrect data. The final processed dataset from the Secondary Resources Survey in Lin'an contains 31 factors, including the land type, area, landform, elevation (m), gradient (degree), slope aspect, slope position, soil name, soil texture, soil-layer thickness (cm), humus-layer thickness, site-quality grade, land-management type, forest land-protection grade, traffic area, forest land-quality grade, undergrowth vegetation type, undergrowth vegetation height (m), undergrowth vegetation total coverage (%), forest category, forest belt length (m), forest belt width (m), number of rows, age, average diameter at breast height (cm), average height (m), canopy density or coverage, density, number of trees (plants) per mu, number of trees (plants), and the afforestation seedling age.

The satellite data were obtained from the geospatial network; the corresponding regional Landsat8 satellite strip data were found and downloaded, and the raster images were transformed to band values and extracted to a CSV file format using Arcgis 10.7 software

(ESRI Inc., Redlands, CA, USA). Then, the data were processed for NULL values and validated for reasonableness, resulting in more than 110,000 data containing 6 optical factors and 2 infrared factors; the bands were B2, B3, B4, B5, B6, B7, B10, and B11.

Finally, the processed data from the Secondary Resources Survey in Lin'an and the Landsat8 satellite data were combined, and the FID of the data of the Secondary Resources Survey and the FID of the satellite data were used as the main keys to merge the two tables using an inner join. The operation was conducted in a Centos 7.9 and MySQL 5.7.36 environment, and the results represent are the final combined data. The final data specification before machine learning was performed on the combined data, which were processed to form the final dataset; this dataset included 39 factors and more than 100,000 rows of dataset.

### 2.3. Machine-Learning Training

After data pre-processing, we needed to select a suitable algorithm to train and learn the data. We attempted to pick a suitable algorithm from each of the three categories of traditional machine learning algorithms, integrated algorithms, and deep learning. We compared a total of five evaluation indicators, using $R^2$, MSE, RMSE, MAE, and SMAPE to filter out the most accurate algorithm to achieve growth-potential predictions based on machine learning.

### 2.3.1. Feature Engineering

The distance correlation coefficient was used for feature selection. If all candidate variables were used to train the model, it would produce redundant information and reduce the interpretability of the model. Therefore, the extracted modeling factors had to be screened. In this study, the distance correlation coefficient was used to measure the correlation between the variables and the stock volume, and independent variables more suitable for estimating the stock volume were screened out. The distance correlation coefficient not only reflects the linear relationship between variables but also expresses the nonlinear relationship between variables; additionally, it does not require any model assumptions or parameter conditions [27]. The 39 factors selected as the characteristics were as follows: land type, area, landform, elevation (m), slope (degree), slope aspect, slope position, soil name, soil texture, soil-layer thickness (cm), humus-layer thickness, site-quality grade, land-management type, forest-land protection grade, traffic area, forest land-quality grade, undergrowth vegetation type, undergrowth vegetation height (m), undergrowth vegetation total coverage (%), forest category, forest belt length (m), forest belt width (m), number of forest belt rows, age, average diameter at breast height (cm), average height (m), canopy density or coverage, density, number of trees (plants) per mu, number of trees (plants), afforestation seedling age, Band2, Band3, Band4, Band5, Band6, Band7, Band10, and Band11.

### 2.3.2. Stochastic Gradient Descent SGD

To address the problem of the regression of many forest stock volumes in traditional machine-learning methods, as described in the official Apache Sklearn documentation, a stochastic gradient descent SGD is recommended [28]. The regression problem requires the use of an SGDRegressor, in which the loss function is a variable parameter; this has a relatively significant impact on the experiment. The loss parameter can be selected from ordinary least squares, the Huber loss for robust regions, and the linear support-vector region. The three parameters are compared in the experiment, and the least- squares method is the best choice. The learning-rate parameter is set relatively low, with eta0 equal to 0.0005, and the other parameters of the SGDRegressor are a validation _fraction equal to 0.9, max_iter equal to 2000, tol equal to 0.00002, and the SGD algorithm, as shown in Figure 3 below.

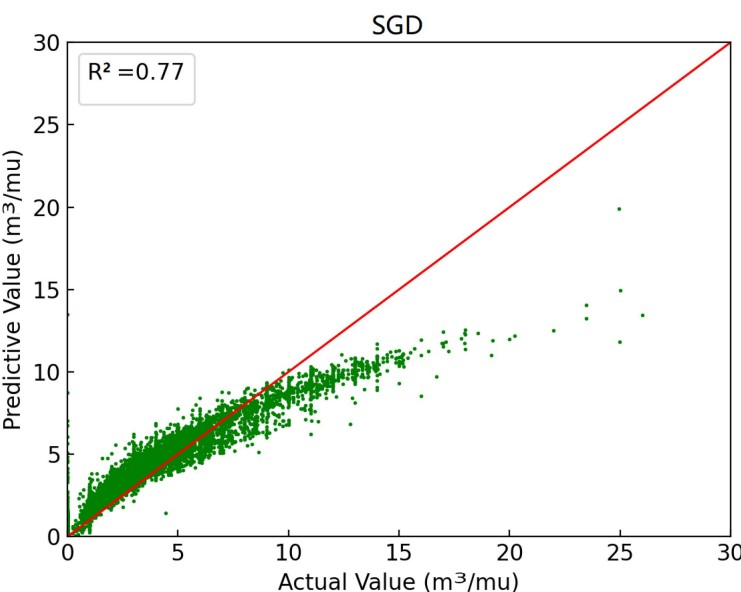

**Figure 3.** Prediction-modeling results of the SGD algorithm.

### 2.3.3. Integrated Machine Learning Algorithm CatBoost

As integration algorithms are the current focus of machine learning, CatBoost, which was positively evaluated, was used. CatBoost is suitable for handling large amounts of data and is well suited to handling the 110,000 data used in this experiment. CatBoost is a new integration algorithm based on decision-tree gradient boosting. It uses a symmetric tree as its base learner and, through a set of base learners in a series iteration, the final strong learner is obtained. The goal of the $k$th iteration of CatBoost is to find $h_k$ to minimize the loss function of this round [29]. Assume that:

$$h_k = argmin_{h \in k} \frac{1}{m} \sum_{k=1}^{m} [-f_k(x_k, y_k) - h(x_k)]^2 \tag{3}$$

In the formula $f_k(x_k, y_k) = \frac{\partial L[y, F_{k-1(x)}]}{\partial F_{k-1(x)}}$, the gradient estimate, $\partial F_{k-1(x)}$, is the current learner formed by the completed $k-1$ iteration, and $\partial L[y, F_{k-1(x)}]$ is the loss function [24].

### 2.3.4. Improved CatBoost Based on Optuna

If the hyperparametric model is not properly adjusted, the performance may be significantly affected, and manual adjustment of the hyperparameter is, labor-intensive. The most common method used in parameter optimization is Grid Search [30], a relatively simple brute-force method. It is important to consider the cost of brute-force searches in terms of the complexity of combining every parameter input into the search space [31]. The second obvious disadvantage of this approach is that the trained model has always performed poorly when used as a large part of the search. Optuna, an optimization tool for hyperparameters, can solve these problems when integrated into the CatBoost model [32]. Optuna optimizes the tree-based hyperparameter search, using a method called the TPE sampler (a tree-structured Parzen estimator); it relies on Bayesian probability to determine which hyperparameter choices are the most promising and to iteratively adjust the search [33]. First, a learning function is established, which specifies the sample distribution of each superparameter. The available options are categorical, integer, float, or log uniform. To set values in the ranges of 0.001, 0.01, and 0.1, where each value has the same probability of being selected, log uniform can be used [34]. Another advantage of Optuna is that it can set conditional superparameters. Many superparameters are effective only when they are combined with other superparameters; changing only them may not produce the

expected effect [34]. The CatBoost model has a very large number of superparameters, so it is appropriate to integrate Optuna to improve the CatBoost model. Optuna gives the optimal superparameters of the CatBoost model, as shown in Figure 4 below.

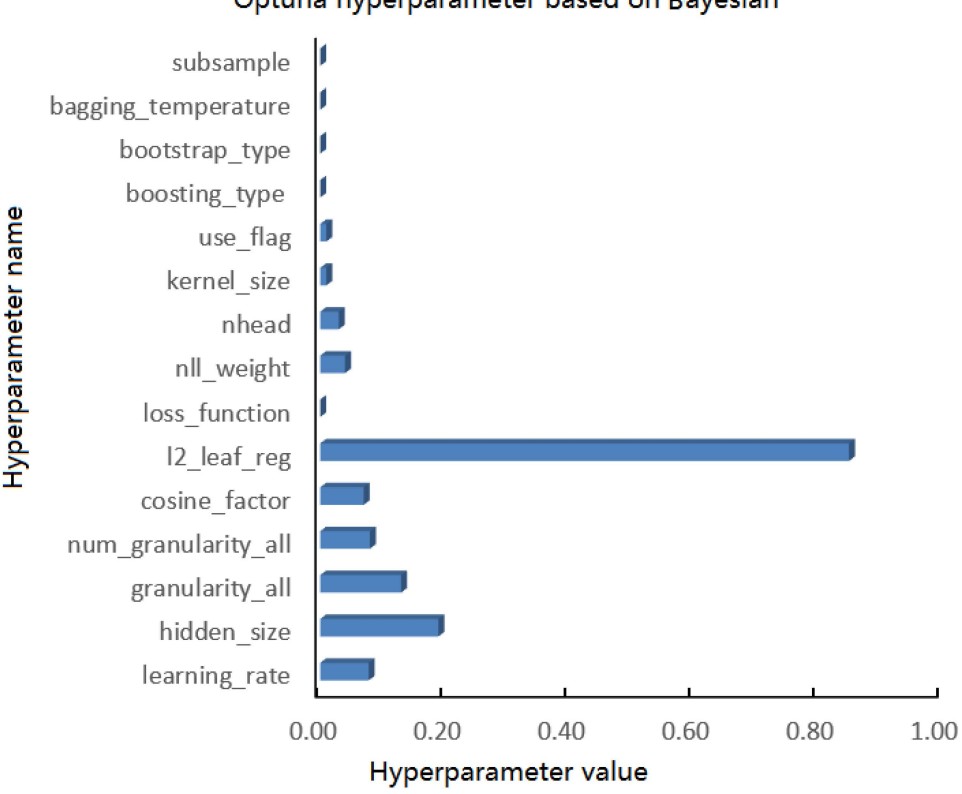

**Figure 4.** Improved CatBoost hyperparameters based on Optuna.

2.3.5. Ablation Experiment

Ablation experiments are commonly used to verify the effectiveness of a certain aspect of the improvement, proving that it is effective [35]. These experiments are borrowed from the medical field and are also commonly used as a basic experimental method in artificial intelligence. Based on the improved CatBoost algorithm model with Optuna, this ablation study is specifically designed to determine the effect of Optuna's Bayesian method for the optimization of superparameters. The comparison includes three parts: CatBoost represents the original algorithm model; CatBoost-Grid represents the CatBoost algorithm model with grid search hyperparametric optimization; and CatBoost-Optuna represents the Catboost algorithm model with Optuna hyperparametric optimization. Figure 5 shows the loss values of three groups of algorithm models: CatBoost, CatBoost-Grid, and CatBoost-Optuna. It can be determined that the maximum loss value of CatBoost is the worst and the minimum loss value of CatBoost-Optuna is the best.

2.3.6. Experimental Effect of Improved CatBoost

CatBoostClassifier is used for classification training, and the prediction of the forest growth-potential value constitutes a data regression problem. Another branch of the CatBoostRegressor algorithm must be adopted in the experimental training model. According to the results of the previous ablation experiments, the loss value of the improved CatBoost model based on Optuna is obviously lower than that of the original model. Although the intelligent optimization of hyperparameters is realized using the Optuna architecture, some parameters still need to be set subjectively. Several important hyperparameters are set manually, including the maximum number of trees, which is set to 100, and the maximum

depth of trees, which is set to 8 layers; the search algorithm adopts a greedy algorithm, and the measure used to detect overfitting is RMSE, as shown in Figure 6 below.

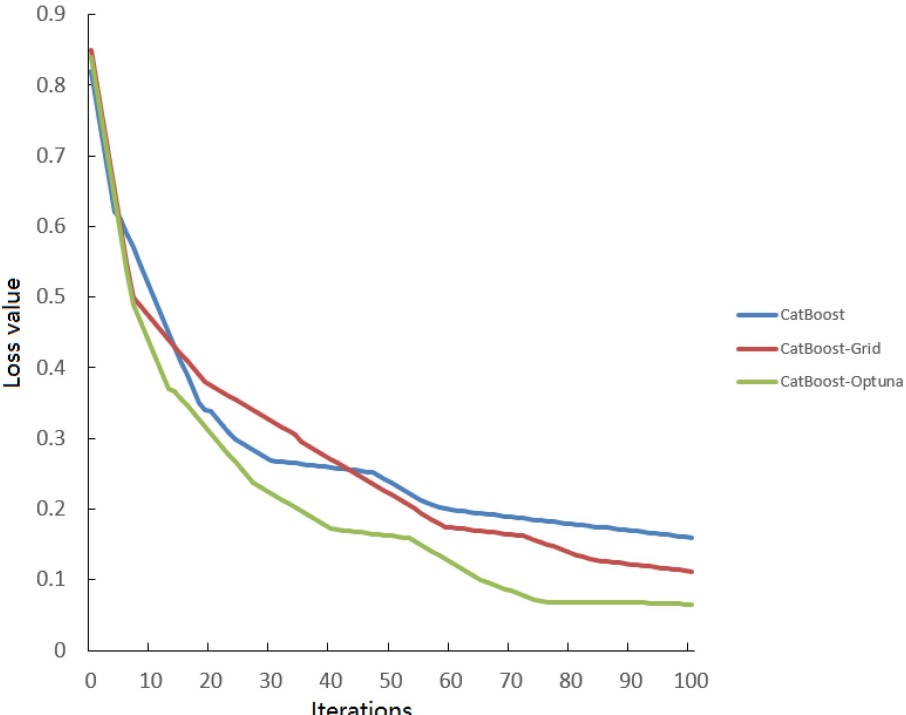

**Figure 5.** Comparison of improved CatBoost methods in ablation experiments.

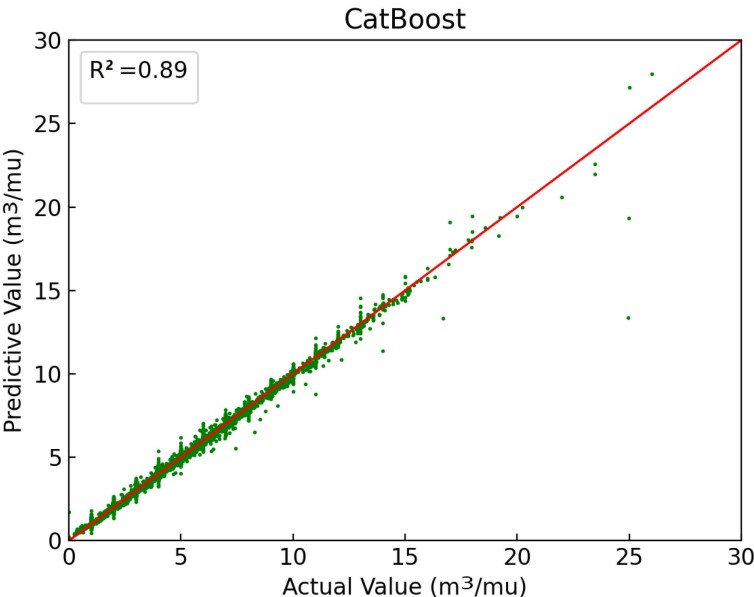

**Figure 6.** Results of CatBoost Algorithm-Prediction Modeling.

### 2.3.7. Deep-Learning CNN

Deep learning is a popular direction in machine learning. Common convolutional neural networks that are mainly used for image detection and target detection include DNN, RNN, GAN, etc. [36–38]. The difference between a convolutional neural network and an ordinary neural network is that the former contain a feature extractor composed of a convolution layer and a subsampling layer (a pooling layer). In the convolution layer of a convolutional neural network, a neuron only connects with some neurons in the adjacent layer. The convolution layer of a CNN (convolutional neural network) usually contains several feature maps [39].

Each feature map is composed of some neurons arranged in a rectangle. The neurons of the same feature map share weight values, and the shared weight values here are convolution cores. This experiment uses a one-dimensional convolutional neural network, with epochs equal to 50 and a batch size equal to 300, as shown in Figure 7 below.

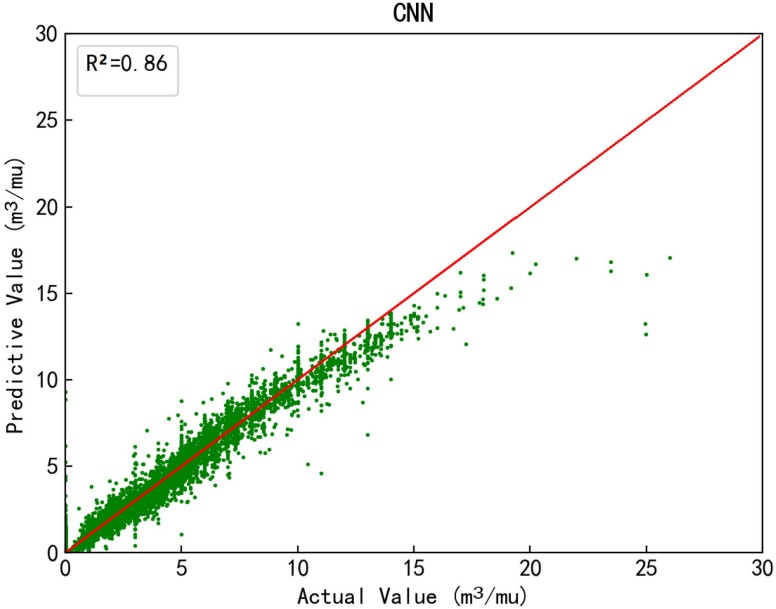

**Figure 7.** CNN algorithm-prediction modeling results.

2.3.8. Evaluating Indicator

A *k*-fold CV divides the dataset into *k* subsets and takes one of the *k* subsets as the validation dataset and the rest of the *k*−1 datasets as the training set. The average of the predicted accuracy of the *k* models calculated is taken as the final accuracy value. This study uses the fivefold cross-validation method. Through the fivefold verification method, the determination coefficient, mean square error, root mean square error, average absolute error, and symmetric average absolute percentage error are compared. Finally, the integrated machine-learning algorithm CatBoost is adopted, with an $R^2$ coefficient of 0.89. The evaluation performance indicators used are R-squared ($R^2$), the mean square error (MSE), the root mean square error (RMSE), the mean absolute error (MAE), and the symmetric mean absolute percentage error (SMAPE) [40–44].

*2.4. Calculating Forest Growth-Potential Value*

SGD is the most basic random gradient descent; it does not require the loss functions for all training data. Instead, it randomly optimizes the loss function of a certain piece of training data in each iteration, so that the updating speed of the parameters in each iteration is significantly accelerated; this effect means that it is not the best choice [45]. The $R^2$ decision coefficient is 0.77, lower than that of the CatBoost algorithm. CatBoost is a gradient-lifting machine-learning algorithm. Gradient-lifting technology usually provides better results than neural networks on heterogeneous datasets, and its algorithm effect is the best. The $R^2$ determination coefficient is 0.89. A convolutional neural network (CNN) is a feedforward neural network. Artificial neurons can respond to the surrounding units and usually perform large-scale image processing. Here, CNN is used for regression tasks and shows good results. The $R^2$ determination coefficient reaches 0.86. Table 3 shows the specific comparison data.

**Table 3.** Comparison of the three machine-learning algorithms.

| Algorithm | Evaluating Indicator | Abbreviation | Value |
|---|---|---|---|
| Random gradient descent SGD | Coefficient of determination | $R^2$ | 0.77 |
| | Mean square error | MSE | 1.58 |
| | Root mean square error | RMSE | 1.26 |
| | Mean absolute error | MAE | 0.88 |
| | Symmetrical mean absolute percentage error | SMAPE | 97.56 |
| Improved CatBoost | Coefficient of determination | $R^2$ | 0.89 |
| | Mean square error | MSE | 0.05 |
| | Root mean square error | RMSE | 0.22 |
| | Mean absolute error | MAE | 0.08 |
| | Symmetrical mean absolute percentage error | SMAPE | 85.98 |
| Deep learning CNN | Coefficient of determination | $R^2$ | 0.86 |
| | Mean square error | MSE | 0.32 |
| | Root mean square error | RMSE | 0.56 |
| | Mean absolute error | MAE | 0.29 |
| | Symmetrical mean absolute percentage error | SMAPE | 153.67 |

In addition to the above five evaluation indexes, R2, MSE, RMSE, MAE, and SMAPE, scatter plots can also reflect the advantages and disadvantages of the model to some extent. A scatter plot can also intuitively show a comparison of the three machine-learning algorithms. The random gradient descent SGD, the integrated algorithm CatBoost, and the convolutional neural network CNN can all be used to predict the unit stock, but the best effect is produced by CatBoost. The scatter plots of the other two algorithms have obviously shifted, as shown in Figures 5–7. The CatBoost algorithm is selected as the machine-learning model for the unit stock prediction, the trained model is saved to a disk, 10 years are added to the following years, and $V_{cat}^{y+10}$ is calculated once again. Then, the corresponding data are substituted into the previous formula (2) to calculate the growth potential.

### 3. Results

*3.1. Application of the Forest Growth Potential Value*

There are 59,027 subcompartments of forest resources in Lin'an District. Overall, most of them were found to have low growth potential, so measures should be taken according to the actual situation of each. A small number of subcompartments have high growth-potential values, and the unit forest volume will increase significantly in the future, which is of positive significance for the double carbon goal [46]. In the land classification system for forest land, there are eight kinds of forest land: arbor, bamboo, sparse, nursery, trail, shrub, undeveloped, and suitable. The forest growth-potential value is only applicable to arbor forest land and sparse forest land.

*3.2. Forest Growth Potential of Arbor Forest Land*

The histogram in Figure 8 below shows the number of subcompartments with a growth-potential value of 5 percentage points. The abscissa is the interval of the forest growth-potential value, and the ordinate is the number of subcompartments of forest resources; the number of subcompartments with a growth-potential value below 5% is 46,336. With the increase in the growth-potential value on the abscissa, the number of subcompartments develops an obvious decreasing trend. The circular chart in Figure 9 below shows the division of the growth-potential grades. Red represents the proportion and number of subcompartments with the lowest forest growth potential, blue represents the proportion and number of subcompartments with medium forest-growth potential, and green represents the proportion and number of subcompartments with the highest forest-growth potential. The red "early warning" section of the growth-potential index is less than 0.85 with 14,702 subcompartments, accounting for 25%. The blue "normal" section refers to a growth potential of more than or equal to 0.85 and less than or equal to 4.37; it comprises 29,424 subcompartments, accounting

for 50%. Finally, the green or "good" section refers to a growth potential of more than 4.37; this section has 14,731 subcompartments, accounting for 25%. A red warning means that the growth potential is low, human intervention is needed, and the increase in future stock is very small. Normal or blue means that the growth potential is medium, and human and material resources are available for further treatment. Good (green) indicates a high growth-potential index; here, the forest stock will increase significantly in the future, so there is no need for major interference with its growth.

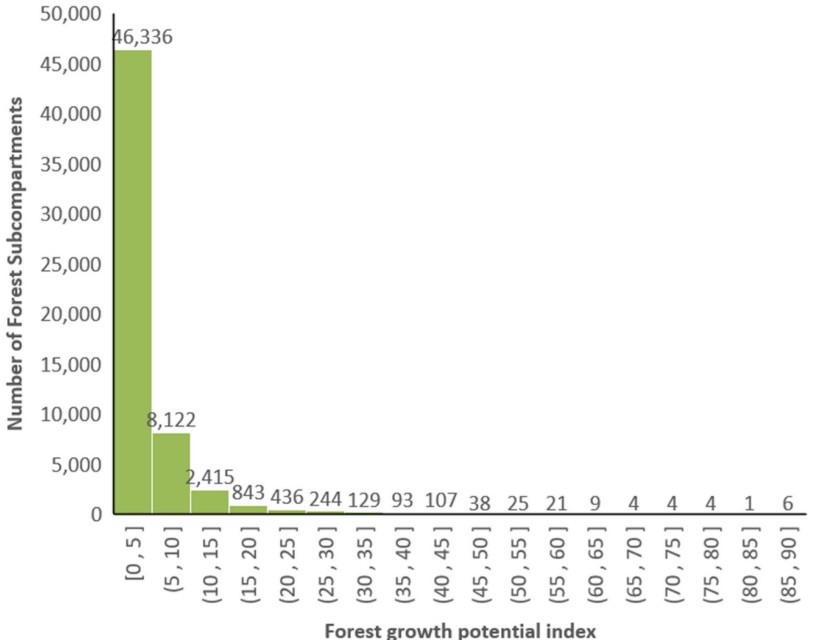

**Figure 8.** Distribution of the forest growth potential of arbor forest land in Lin'an District.

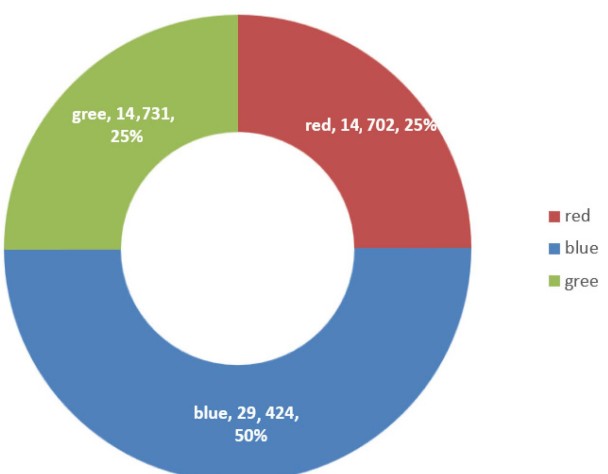

**Figure 9.** Grade of the forest growth potential of arbor forest land in Lin'an District.

### 3.3. Forest Growth Potential of Sparse Forest Land

In the histogram in Figure 10 below, the abscissa is the interval of the forest growth-potential value, and the ordinate is the number of subcompartments of forest resources. The forest growth-potential values for sparse woodlands are all normal, without removing the 10 maximums and minimums; this shows the number of subcompartments with growth-potential values for every 5 percentage points, and the number of subcompartments with growth-potential values below 5% is 49. With the increase in the growth-potential values on the abscissa, the number of subcompartments exhibits a trend of stability at the front and a decrease at the back, indicating that the growth-potential values of sparse

woodlands are higher than those of arbor woodlands. The ring diagram in Figure 11 below shows the division of the growth-potential grades. Red represents the proportion and number of subcompartments with the lowest forest growth potential, blue represents the proportion and number of subcompartments with medium forest growth potential, and green represents the proportion and number of subcompartments with the highest forest growth potential. The number of subcompartments with a growth potential index of less than 5.59, in red, is 14,702, accounting for 25%. The number of subcompartments with a growth potential greater than or equal to 5.59 and less than or equal to 19.28 (blue) is 107, accounting for 50%, and the number of subcompartments with a growth potential greater than 19.28 (green) is good, accounting for 25%. A red warning means that the growth potential is low and human intervention is needed, and the increase in future stock is very small. Normal (blue) means that the growth potential is medium, and there are human and material resources available for further treatment. Good (green) indicates a high growth potential index, meaning that the forest stock will increase significantly in the future, so there is no need for major interference with its growth.

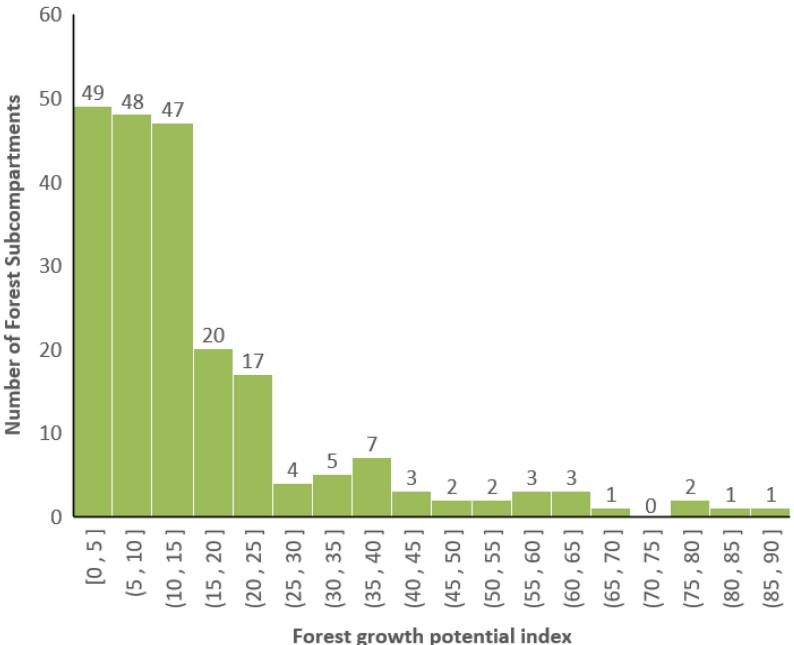

**Figure 10.** Distribution of the forest growth-potential of sparse forest land in Lin'an District.

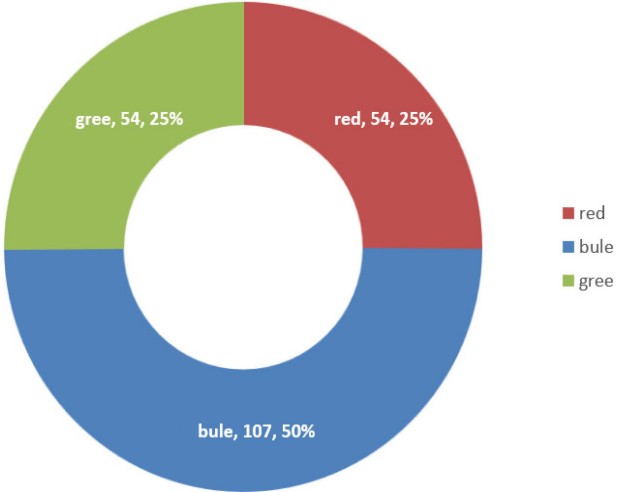

**Figure 11.** Grade of the forest growth potential of sparse forest land in Lin'an District.

## 4. Discussion

### 4.1. Comparison with Other Studies

The core basis of the calculation of growth potential is to predict the unit stock. Han, Rui et al. provide 4000 rows of data for forest stock estimations based on the Boruta and extreme random tree methods [47]. Li, Kun et al. present 40,000 rows of data for forest stock estimation using Sentinel-2 remote-sensing image data [48]. This paper uses more than 100,000 data to train the model. Compared with the calculation accuracy of the stock volume, the data volume has increased significantly, but the prediction accuracy has not changed much, as shown in Table 4 below.

**Table 4.** Comparison with other studies.

| Studies | Year | Geographical Area | Number of Data | $R^2$ |
| --- | --- | --- | --- | --- |
| Estimation of forest stock based on the Boruta and extreme random tree methods | 2020 | Longquan city | 4002 | 0.92 |
| Estimation of forest stock using Sentinel-2 remote sensing image data | 2021 | Linhai city | 19,840 | 0.77 |
| Estimation of forest stock using Sentinel-2 remote sensing image data | 2021 | Chun'an area | 40,216 | 0.83 |
| This article | 2023 | Lin'an city | 111,435 | 0.89 |

### 4.2. The Main Significance of Forest Growth Potential

To some extent, forest growth potential reflects the basic and future quality of forests, and it also characterizes future trends in forest quality, aiming to form a predictive and comprehensive forest quality indicator. By calculating the forest growth-potential value, it is possible to accurately identify which subcompartments have little potential for the forest stock to be improved; these require our focus and intervention. This metric is also highly significant for ecological balance and environmental protection. The forest growth-potential value can quickly and accurately issue warnings in relevant forestry systems, effectively improving forestry intelligence, achieving the precise improvement of forest quality, and applying the data to smart agriculture.

### 4.3. Design Ideas for Forest Growth Potential

Here, we detail some design ideas for forest growth-potential values. Having gained popularity in recent years, machine learning has successfully been applied in multiple fields, and some researchers have attempted to use it to determine the potential value of forest growth. The experimental data consist of the subcompartment data of 110,000 forest resources in Lin'an District and NASA Landsat-8 remote sensing satellite data; forest data are only authorized for the year 2019 in Lin'an District. To fully explore the potential of combining forestry Secondary Resources Survey data with remote sensing data when estimating forest unit volume, not only were three different types of algorithm models designed, but improvements were also made to the CatBoost algorithm model, which produced the best results. After comparing the evaluation indicators, the improved CatBoost model was shown to have the best prediction effect for the forest unit volume, and it also has certain advantages compared with other scholars' machine learning predictions of forest volume. Our method for the prediction of forest unit volume is designed to calculate the forest growth-potential value as accurately as possible. The predicted forest unit-volume value is calculated as a whole in the forest growth-potential value formula, and the forest growth-potential value is ultimately calculated.

## 5. Conclusions

This study investigated the research status of forest quality and accurate forest quality improvements both in China and abroad. Referring to the Secondary Resources Survey forestry regulations, we summarized the forest quality standards of domestic and foreign institutions and experts, successfully finding an index that could predict the future state and

trends of forest quality. We put forward a reasonable new forest-quality index, the forest growth-potential value, and constructed a model for calculating the forest growth-potential value [49]. Three different machine learning algorithm model experiments were completed. We compared the traditional machine-learning random gradient descent SGD, the gradient-lifting CatBoost integrated with machine learning, and the convolution neural-network CNN model of deep learning [50]. We realized the optimization of the CatBoost algorithm model based on Optuna. The CNN algorithm model was adapted for the data regression experiment, and the improved CatBoost model calculated the forest growth-potential value. We completed the calculation, analysis, and application of the forest growth-potential values of nearly 60 thousand subcompartments of forest resources in Lin'an District. The forest growth-potential value, based on the machine learning of subcompartments, can be used for accurate forest quality improvement.

The CatBoost machine learning model has a positive effect on the prediction of the unit stock volume of large quantities of data. At the same time, compared with other machine learning methods, CatBoost is suitable for processing large quantities of data without its accuracy decreasing and has low requirements for data quality. To further increase the accuracy of the growth-potential index, improving the prediction accuracy of the unit stock is key. Whether by increasing the number and diversity of samples or by optimizing the model algorithm, the prediction accuracy of the growth-potential index can be improved.

It is foreseeable that, in the future, with further in-depth research and the availability of richer samples, the division of red, blue, and green could produce a more reasonable division standard. The proportion of arbor forest land in all the datasets is too high, which shows that the growth-potential value of arbor forest land is low; therefore, it is necessary first to undertake forest tending, to give priority to timber cutting, to consider cutting into a non-contiguous state in view of forest fires and forest pests, to consider changing the forest ecology to other tree species, or to consider reducing the number of trees per mu to implement thinning management.

Further increasing the amount of data to enhance universality can produce more datasets in different regions and climatic environments. For instance, it would provide data for different regions of the country, different climatic conditions around the world, different tree species, and scenarios where different human activities affect forests. Another research direction involves an independent algorithm model under multi-classification according to climate or tree species to further increase accuracy and practicability; it might involve subtropical broad-leaved forests, warm temperate-zone coniferous forests, mid-temperate-zone coniferous forests, etc. Each classification realizes the optimal algorithm model independently, and different classifications have different algorithm models.

**Author Contributions:** Conceptualization, L.C. and L.F.; methodology, L.C. and X.H.; software, L.C.; validation, S.C. and L.F.; formal analysis, L.C.; investigation, X.H. and S.C.; data curation, L.C.; writing—original draft preparation, L.C.; writing—review and editing, L.C. and X.H.; visualization, L.C.; supervision, L.C. All authors have read and agreed to the published version of the manuscript.

**Funding:** This work was financially supported by the Zhejiang Provincial Key Science and Technology Project, Grant No. 2018C02013; the National Natural Science Foundation of China, Grant No. 42001354; and the Natural Science Foundation of Zhejiang Province, Grant No. LQ19D010011.

**Institutional Review Board Statement:** Not applicable.

**Informed Consent Statement:** Not applicable.

**Data Availability Statement:** The remote sensing data of the Landsat8 satellite can be found here: [http://www.gscloud.cn/search (accessed on 19 February 2023)]. The first dataset has the data ID LC81190392019313LGN00, strip no. 119, line no. 39, from 9 November 2019. The second dataset has the data ID LC81200392019304LGN00, strip no. 120, line no. 39, from 31 October 2019. The Forestry Secondary Resources Survey data are not publicly available as, for policy reasons, they are kept confidential.

**Conflicts of Interest:** The authors declare that they have no known competing financial interests or personal relationships that could have appeared to influence the work reported in this paper.

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
