# Peer review of "Assessing Forest Quality through Forest Growth Potential, an Index Based on Improved CatBoost Machine Learning"

_sustainability, doi:10.3390/su15118888_

Round 1

Reviewer 1 Report

This study proposed a forest quality index-forest growth potential value model which provides guidance for the accurate improvement of forest quality and forest management. Based on the data of 110,000 Subcompartments of forest resources and Landsat8 satellite remote sensing data, the unit volume is predicted by three machine learning algorithms: random gradient descent SGD, integrated machine learning algorithm CatBoost and deep learning CNN. The CatBoost algorithm model is improved based on Optuna, and then the model is constructed, the forest growth potential value is calculated, and the accurate forest quality improvement scheme based on Subcompartments is preliminary discussed. The improved CatBoost calculation model has a good effect on the prediction of forest growth potential value. 

The paper can be accepted after addressing the following issues.

Paper structure:

the entire structure of this paper should be improved. The authors can refer to a structure of Introduction, Material and methods, Results, Discussions, Conclusion.

Abstract:

--What are the highlights or innovation of this study ? The authors should clarify.

--This part can be refined to improve its readability and attraction.

--What is the meaning of Subcompartment ?

Introduction:

--The authors should refine this section to clarify the study trend of forest growth potential and commonly used methods, and show the main issue and gap that the study is going to address.

Research Area and Data Source

--A figure that shows the location of the study area is needed in this section.

Forest Quality Indicators

--Table 2, the reference citation should be added to the end of each source.

Experimental process

--This section can be reconstruct to make it more clear.

--Figure 6 and 7, R2 should be R2.

Discussions

--There should be a section of Discussions.

--The innovation should be highlighted in this section.

Moderate editing of English language is recommended.

Author Response

Thank you for the valuable sugestion,and I have made all possible modification. Please refer to the attached document.

Reviewer 2 Report

Dear Authors,

Thank you for submitting your article entitled “A New Way of the Forest Quality of Forest Growth Potential Based on Improved CatBoost Machine Learning” for consideration. While your paper has some interesting aspects, I regret to inform you that I cannot accept it for publication at this time.

One of the main concerns I have is the lack of clarity in the manuscript's flow and syntax, making it difficult to follow the logic of your scientific writing. There are also issues with the organization of the manuscript, where the Results and Materials and Methods sections are blended together. This makes it unclear where the methods start and where the results begin.

Additionally, there are several instances throughout the text where relevant references are missing, and the Discussion section is completely missing. The discussion section is crucial in highlighting the significance of your findings in comparison with similar studies, as well as discussing its limitations.

Furthermore, I would like to point out that the article appears to be a technical report rather than a scientific article!!!

I strongly suggest that you work on addressing these issues before submitting your paper for publication again. It is important to ensure that your manuscript is well-organized, the syntax and flow are clear, and that all necessary references are included. Please note that the quality of writing is essential in scientific publications, and it is crucial to ensure that the manuscript is well-written and easy to follow.

Thank you for considering these comments

Introduction

1.       The introduction section is way too short. You should increase its length.

2.       Lines 35-38: Please rephrase better. If you want follow my proposal: “The forest is a prominent carbon sink in the terrestrial ecosystem, serving as a crucial mechanism for mitigating the greenhouse effect. Furthermore, the forest functions as a vital reservoir for terrestrial resources, including water and soil conservation. As an ecological barrier, the forest serves an essential role in curtailing the adverse effects of sandstorms and minimizing land desertification.”

3.       Lines 39-40: Here you can start a new paragraph after the 2nd reference.

4.       Line 40: Round the number of hectares from 158.9409 to 159

5.       Lines 50-53: I suggest using active voice (“we used a variety of machine learning models…)

6.       Lines 50-53: Write in more detail the aims of your study as well as research questions.

Materials and Methods

7.       Line 55: This section is entitled Materials and Methods. Please correct the title. After this title, you can add the sub-title “Research Area and Data Source”.

8.       Line 58-59: I do not understand the term “clear stereo climate”. Please correct the mistake!

9.       Lines 62: Here you can start a new paragraph after the 8th reference.

10.   Lines 65-66: Reformulate the sentence.

11.   Line 83: This report is 8 years old. You should use FAO’s last report (2020) (check the following link: https://www.fao.org/documents/card/en/c/ca9825en)

12.   Line 83: I could not find the indicators that you describe as “landscape indicators”. Please provide me with more information (the exact page of the report).

13.   Line 88: instead of the state use the actual name of it, China!

14.   Line 94-95: Re-formulate the sentence by removing the names of the authors (follow this instruction in all the text, lines 103, 127 etc.).

15.   Line 95: The reference that you cited is not in Chinese as you stated (Feng, J.; Wang, J.; Yao, S.; Ding, L. Dynamic assessment of forest resources quality at the provincial level using AHP and cluster analysis. Comput. Electron. Agric. 2016, 124, 184–193)

16.   Line 103: The 3.1 section does not contain any analytical process, is just a literature review related to forest indices. So, remove the term “analysis”.

17.   Lines 72-133: All of this text should be moved in the Introduction section. Moreover, in Table 2 you should add a column with the year of publication (for each row).

18.   Lines 135-147: This paragraph has no logical flow in relation to the previous one. Some parts of this text belong to the Introduction section and in particular to the aims of your article.

19.   Lines 142-147: The equation does not fit in this part of the text.

20.   Lines 150-155: This part should be moved to the Introduction section as it is described clearly your aim.

21.   Lines 156: “and its R2 coefficient is as high 154 as 0.89” This is a result!!!!!!!!

22.   Line 157: You stated that: “In the short and medium term, 10 years is an ideal time period”. You should find some relevant and up-to-date scientific literature to prove it!!!!

23.   Lines 169-171: Describe the methods that you “clean” the raw data.

24.   Lines 171-173: Add reference.

25.   Line 176: Describe the 31 final factors of the model or add a table.

26.   Line 179: Add reference inside the written text for the ArcGIS software (ESRI Inc., Redlands, CA, USA).

27.   Lines 181: Describe or just mention the optical and the infrared factors.

28.   Line 192-193: five evaluation indicators

29.   Lines 201-203: Add reference!!!

30.   Lines 209: I cannot understand why the number of rows is a variable contained in this dataset!

31.   Lines 211-212: B2, 211 B3, B4, B5, B6, B7, B10, B11????????????Explain!!!!

32.   Line 216-217: Who did the comparison???

33.   Line 221: The R square in the figure has to be superscript (R2 = 0.77).

34.   Lines 238, 240, 241, 246: Add reference! Avoid use the word unacceptable!

35.   Line 256: The Figure 3 is not entirely correct. You should add labels in both axes and you should explain in more detail the superparameters of the model.

36.   Lines 259-261: You have to explain in more detail this part.

37.   Lines 268: Figure 4 is not entirely correct. You should add labels in both axes and you should explain in more detail the outcome.

38.   Line 282: The R square in the figure has to be superscript (R2 = 0.89).

39.   Lines 287-293: Add references

40.   Lines 310-314: The equations of the evaluation indices are well known, so if you want you can exclude them from the text.

41.   Line 318: Add reference

42.   Line 319: You stated that: “The R2 decision coefficient is only 0.77”. R2 is the coefficient of determination. The R square is a number that measures how well a model predicts an outcome. In this case, a prediction of 77% is not so low, so re-write the above sentence!

43.   Line 338-346: This should be moved to the Discussion section

44.   Line 346: Table 4 needs a column with the year of each study, and the title of each one not the names of the first authors!

Results

45.   Lines 347: This section is entitled Results. Please correct the title. After this title, you can add the sub-title “Application of forest growth potential value”.

46.   Lines 372, 374: Figures 8 and 9 should explain in more detail the outcome

Discussion

47.   Unfortunately Discussion section is missing. You should definitely discuss your findings by writing the importance of the detection algorithm as well as its limitations. Despite the fact that this work in general is interesting the absence of Discussion is crucial.

Conclusions

48.   You have to add some references!!!

References

49.   Follow the authors' guidelines of the journal!

The manuscript had some significant issues with grammar, syntax, and spelling, which made it difficult to follow scientific writing. It is essential to ensure that your manuscript is grammatically correct and written in clear and concise English to allow for a proper understanding of your research.

I would recommend that you seek the help of a professional English language editor to revise your manuscript before submitting it for consideration. This will ensure that your paper is written in appropriate academic English, and the content can be easily understood by your target audience.

Author Response

(The authors gave the same response as above.)

Round 2

Reviewer 2 Report

Dear authors,

Thank you for your effort to improve your manuscript. This is a much better, improved, and well-written version that deserves to be published!

I have just three minor issues that need to be corrected:

1.       Lines 72: Correct the missing word

2.       Lines 84: Correct the sentence.

3.    Line 202: The location of the study area should be in English not in Chinese (cities, towns etc.). 

Author Response

Thanks for your careful and reasonable advice, which has greatly improved the readability and value of this article. Please refer to the attachment for details.
